# Pathological Role of Pin1 in the Development of DSS-Induced Colitis

**DOI:** 10.3390/cells10051230

**Published:** 2021-05-17

**Authors:** Yasuka Matsunaga, Shun Hasei, Takeshi Yamamotoya, Hiroaki Honda, Akifumi Kushiyama, Hideyuki Sakoda, Midori Fujishiro, Hiraku Ono, Hisanaka Ito, Takayoshi Okabe, Tomoichiro Asano, Yusuke Nakatsu

**Affiliations:** 1Department of Medical Chemistry, Division of Molecular Medical Science, Graduate School of Biomedical Sciences, Hiroshima University, 1-2-3 Kasumi, Minami-ku, Hiroshima 734-8553, Japan; ymatsunaga@tulane.edu (Y.M.); b151007@hiroshima-u.ac.jp (S.H.); ymmty@hiroshima-u.ac.jp (T.Y.); 2Research Fellow of Japan Society for the Promotion of Science, Tokyo 102-0083, Japan; 3Department of Disease Model, Research Institute of Radiation Biology and Medicine, Hiroshima University, Hiroshima 734-8553, Japan; honda.hiroaki@twmu.ac.jp; 4Field of Human Disease Models, Major in Advanced Life Sciences and Medicine, Institute of Laboratory Animals, Tokyo Women’s Medical University, Tokyo 162-8666, Japan; 5Department of Pharmacotherapy, Meiji Pharmaceutical University, Kiyose City, Tokyo 204-8588, Japan; kushiyama@my-pharm.ac.jp; 6Division of Neurology, Respirology, Endocrinology and Metabolism, Department of Internal Medicine, Faculty of Medicine, University of Miyazaki, Miyazaki 889-1692, Japan; hideyuki_sakoda@med.miyazaki-u.ac.jp; 7Division of Diabetes and Metabolic Diseases, Department of Internal Medicine, Nihon University School of Medicine, Tokyo 173-8610, Japan; fujishiro.midori@nihon-u.ac.jp; 8Department of Endocrinology, Hematology and Gerontorogy, Graduate School of Medicine, Chiba University, 1-8-1 Inohana, Chuo-ku, Chiba 260-8670, Japan; hono@chiba-u.jp; 9School of Life Sciences, Tokyo University of Pharmacy and Life Sciences, 1432-1 Horinouchi, Hachioji, Tokyo 192-0392, Japan; itohisa@toyaku.ac.jp; 10Drug Discovery Initiative, The University of Tokyo, 7-3-1 Hongo, Bunkyo-ku, Tokyo 113-0033, Japan; zokabety@g.ecc.u-tokyo.ac.jp

**Keywords:** Pin1, ulcerative colitis, dextran sodium sulfate, Pin1 inhibitor, knockout mice

## Abstract

Inflammatory bowel diseases (IBDs) are serious disorders of which the etiologies are not, as yet, fully understood. In this study, Peptidyl-prolyl cis-trans isomerase NIMA-interacting 1 (Pin1) protein was shown to be dramatically upregulated in the colons of dextran sodium sulfate (DSS)-induced ulcerative colitis model mice. Interestingly, *Pin1* knockout (KO) mice exhibited significant attenuation of DSS-induced colitis compared to wild-type (WT) mice, based on various parameters, including body weight, colon length, microscopic observation of the intestinal mucosa, inflammatory cytokine expression, and cleaved caspase-3. In addition, a role of Pin1 in inflammation was suggested because the percentage of M1-type macrophages in the colon was decreased in the *Pin1* KO mice while that of M2-type macrophages was increased. Moreover, *Pin1* KO mice showed downregulation of both *Il17* and *Il23a* expression in the colon, both of which have been implicated in the development of colitis. Finally, oral administration of Pin1 inhibitor partially but significantly prevented DSS-induced colitis in mice, raising the possibility of Pin1 inhibitors serving as therapeutic agents for IBD.

## 1. Introduction

Inflammatory bowel diseases (IBDs), including Crohn’s disease (CD) and ulcerative colitis (UC), are serious disorders. The pathophysiology underlying these diseases has yet to be fully clarified. IBDs are chronic relapsing disorders characterized pathologically by inflammation and epithelial injury involving the gastrointestinal tract. Symptoms include abdominal pain, diarrhea, rectal bleeding, weight loss and fatigue in both Crohn’s disease and ulcerative colitis [1,2].

Recent studies have identified a major role of both genetic and environmental factors in the pathogenesis of IBD. Genetic and environmental risk factors trigger epithelial barrier function deterioration [3,4]. Impaired barrier function results in the invasion of commensal bacteria from the gut lumen into the bowel wall, which then leads to immune cell activation and cytokine production [5,6]. If acute mucosal inflammation cannot be resolved by anti-inflammatory mechanisms, chronic and repetitive intestinal inflammation develops. Chronic inflammation ultimately leads to disease complications as well as tissue destruction, both of which are driven by mucosal cytokine responses.

In this study, first, we found that Pin1 protein was dramatically upregulated in the colons of dextran sodium sulfate (DSS)-induced colitis model mice, and therefore, investigated Pin1 involvement in the pathogenesis of IBD. Pin1 is a unique peptidyl-propyl isomerase, which specifically recognizes phosphorylated serine or threonine residues located at the immediate N-terminus to a proline, followed by isomerization of the bound peptide and catalyzation of the isomerization of prolyl peptide bonds from trans to cis. These conformational changes stabilize proteins, protein–protein interactions, cellular localizations and catalytic activities, responses which vary according to the individual protein. As a consequence, Pin1-catalyzed conformational changes have a profound impact on many key proteins playing roles in the regulation of cell growth, immune responses, differentiation and survival [7,8,9,10]. Notably, Pin1 binds to Nuclear factor kappa B (NF-κB), an important transcription factor regulating inflammation and immune processes, and stimulates its transcriptional activity [10].

Pin1 involvement has been identified in numerous biological processes related to pathological conditions, including cancer, inflammation, Alzheimer’s disease, aging, asthma and microbial infection [10,11,12,13]. To our knowledge, this is the first study to unravel the roles of Pin1 in inflammatory diseases, notably colitis, and raise the possibility of Pin1 inhibitors serving as novel therapeutic agents for IBD.

## 2. Materials and Methods

### 2.1. Animal

The generation of *Pin1* fl/fl mice on a C57BL/6 background was reported previously and is, thus, not repeated in detail [14]. To create Pin1 null mice, *Pin1* flox mice were crossed with CAG-Cre mice. The genotyping primers used were P1 (tct cct tcc act ggg caa ctt cct g) and P2 (tgc tgt ccc cat cgg gac ct) for both the wild and the floxed type allele, and P1 and P3 (ctg tgg tgc tct tgg ggg tg) for the knockout (KO) allele, yielding 180 bp, 375 bp, and 419 bp products, respectively. Animal studies were conducted using age-matched controls for each experiment. Eight- and 10-week-old mice were used for the experiments. Colitis was induced in these mice by administering 3% DSS (molecular weight 40,000; Wako, Osaka, Japan) in drinking water for 7 days.

For treatment with Pin1 inhibitor, juglone was dissolved in dimethyl sulfoxide (DMSO) and orally administered to the mice (5 mg/Kg BW).

These experiments were approved by Hiroshima University Animal Research Committee. The animals were handled in an accordance with the animal experimentation guidelines of Hiroshima University (approval number A15-1).

### 2.2. Antibodies

Anti-Pin1 (sc-46660, RRID: AB_628132), anti-α tubulin (sc-8035, RRID: AB_628408) and anti-actin (sc-8432, RRID: AB_626632) antibodies were purchased from Santa Cruz Biotechnology (Dallas, TX, USA). Anti-active caspase-3 (9661, RRID: AB_2341188) antibody was purchased from Cell Signaling (Boston, MA, USA).

### 2.3. Flow Cytometry

Flow cytometry analysis was performed using LSRFortessa X-20 (BD Biosciences, Franklin Lakes, NJ, USA). Anti-PE-F4/80 (eBioscience, Waltham, MA, USA), anti-APC-CD11c and anti-Brilliant Violet 421-CD206 (BioLegend, San Diego, CA, USA) were used for the flow cytometric analyses.

### 2.4. Real Time PCR

Real time PCR was performed, as previously reported [14,15]. Briefly, total RNA from tissues including the intestines was isolated using Sepazol reagent (Nacalai Tesque, Kyoto, Japan). We used an Oligotex™ -dT30 mRNA Purification Kit to isolate mRNA from total RNA obtained with Sepazol. For every sample, the 260/280 Absorbance Ratio was near 2.0. To obtain first-strand cDNA, 50 ng of mRNA were reverse-transcribed using a Verso cDNA Synthesis Kit (Thermo Scientific, Waltham, MA, USA), according to the manufacturer’s instructions. This kit contains an RT enhancer reagent which prevents the contamination of genomic DNA. Real-time PCR was performed using the CFX96 real-time PCR system (Bio-Rad, Hercules, CA, USA) with SYBR Premix Ex Taq (Takara-Bio, Kusatsu, Japan). The data obtained were normalized by the expression level of *Gapdh*. Primer sequences are presented in Table 1.

### 2.5. Histology

Formalin-fixed and paraffin-embedded murine colon tissue was sectioned, followed by standard hematoxylin and eosin (H&E) and Periodic acid-Schiff (PAS) staining. Briefly, deparaffinized sections were stained with hematoxylin. After being washed, they were also incubated with eosin solutions. For PAS staining, deparaffinized slides were reacted with periodic acid. After being washed, they were incubated with Schiff’s reagent and then reacted with hematoxylin. Finally, the slides were dehydrated and embedded.

### 2.6. Immunohistochemical Analysis

For immunofluorescence staining, the entire colon was fixed in 10% formalin and embedded in paraffin. Paraffin-embedded samples were sectioned. The sections were incubated in 0.1% Triton solution. After washing with phosphate buffered saline (PBS), the sections were boiled in citrate buffer (pH = 6.0) followed by a 30-min microwave exposure for antigen retrieval. The slides were then reacted with Pin1 antibody (1:200) at 4 °C overnight. After being washed with PBS, the slides were stained by the avidin–biotin complex staining method, according to the manufacturer’s protocol (Santa Cruz). The samples were examined using a BZ 9000 All-in-one Fluorescence Microscope (Keyence, Osaka, Japan).

### 2.7. Western Blotting

Western blotting was conducted as reported previously [14,15]. Colons were homogenized in lysis buffer containing 50 mM Tris-HCl (pH 7.4), 150 mM NaCl, 1 mM ethylenediaminetetraacetic acid (EDTA), 1% Triton X-100, 1 mM NaF, 1 mM Na_3_VO_4_, and 1 mM phenylmethylsulfonyl fluoride. The lysates were incubated on ice for 30 min and then centrifuged at 15,000 rpm for 10 min and subsequently for 30 min at 4 °C. After adjusting the protein concentrations, the supernatants were mixed and boiled with sample buffer. The same amounts of samples (40 µg) were electrophoresed with SDS (sodium dodecyl sulfate)-polyacrylamide gel, and then transferred to polyvinylidene difluoride membranes. The membranes were blocked with 3% bovine serum albumin for 1 h at room temperature, and then incubated with primary antibody (1:3000). After being washed with PBS containing 0.1% Tween 20 three times, the membranes were reacted with HRP-conjugated secondary antibody (1:5000) for 1 hr at room temperature. After washing, proteins on the membranes were subjected to immunoblotting using Supersignal West Pico Substrate (Thermo Scientific, Waltham, MA, USA).

### 2.8. Statistical Analysis

Data were analyzed in GraphPad Prism (GraphPad Software, La Jolla, CA, USA). For multiple group comparisons, we used a one-way ANOVA with a Tukey multiple comparison posttest to calculate *p* values. The *p* values ≤0.05 are considered to indicate a statistically significant difference.

## 3. Results

### 3.1. Pin1 Expression Is Increased in DSS-Induced Colitis

It is reported that mice given drinking water supplemented with 3% DSS for 7 days show symptoms including diarrhea, rectal bleeding, reduction in colon length, body weight loss and decreased activity, which mimic the clinical and histological features of IBDs. Microscopic observation indicated severe damage in the epithelial layer. Interestingly, Pin1 protein expression levels in the colons of mice treated with DSS for 7 days were dramatically increased, i.e., by approximately 45 folds, as compared with those of control mice, while Pin1 mRNA levels were not significantly altered (Figure 1A,B). Next, we carried out immunostaining using anti-Pin1 antibody, which confirmed elevated Pin1 content in the DSS-treated mouse colons (Figure 1C).

### 3.2. Global Pin1 KO Mice Were Resistant to DSS-Induced Colitis Development

Since Pin1 protein was markedly increased in the colonic tissues of DSS-treated mice, we speculated that Pin1 is involved in the development of colitis. To address this possibility, first, global Pin1 KO mice, generated by crossing Pin1 flox and CAG-Cre mice, were subjected to experiments resulting in the development of DSS-induced colitis. As expected, Pin1 expressions were fully deleted in the colons of KO mice (Appendix A). After 7 days of DSS treatment, the animals were weighed, colon length was measured and microscopic observations were performed. While marked body weight loss occurred in the normal mice treated with DSS for 7 days, Pin1 KO mice showed only slight weight loss (Figure 2A). The colon length reduction was also milder in the Pin1 KO than in the control mice (Figure 2B). The microscopic observations of colonic tissues also showed severe tissue destruction in response to DSS treatment in normal mice, while the severity of these changes was significantly attenuated in Pin1 KO mice (Figure 2C).

In the colons of DSS-treated mice, the expressions of inflammatory cytokines such as *Tnfα*, *Il6* and *Il1b* and the chemokine *Mcp1* were markedly elevated, and also showed significant normalization in the Pin1 KO mice (Figure 2D). In addition, active caspase 3, showing the presence of apoptotic cells in the epithelial layer, was markedly increased in the colonic tissues of DSS-treated normal mice, an increase which was also attenuated in the Pin1 KO mice (Figure 2E). These datasets indicated that deletion of the pin1 gene ameliorates DSS-induced colitis development. In other words, Pin1 plays an essential role in colitis development.

### 3.3. Knockout of the Pin1 Gene Increases the M2 Macrophage Population and Reduces Expression of Th17 Cell-Related Cytokines in the Colons of DSS-Treated Mice

In IBD patients and in mice treated with DSS, pro-inflammatory and anti-inflammatory cytokines are produced by the mucosal immune system in response to environmental triggers. M1 macrophages express pro-inflammatory cytokines. In contrast, M2 macrophages function in wound healing and tissue repair, and turn off damaging immune system activation by producing anti-inflammatory cytokines [16]. Thus, we investigated the populations of macrophage subclasses in WT and Pin1 KO mouse colons. The M1 macrophage population increased, while that of M2 macrophages was reduced, in WT mice treated with DSS (Figure 3A,B). In contrast, DSS-treated Pin1 KO mice showed reduced M1 and increased M2 macrophages, respectively, as compared with WT mice.

Many macrophage populations are normally exposed to sufficient levels of tissue-derived M-CSF to maintain them in an M2 state. During inflammation, exposure of macrophages to GM-CSF leads to a pro-inflammatory M1 state. Expression of *Gmcsf* was markedly increased in WT mice treated with DSS. In contrast, *Mcsf* expression was markedly increased in Pin1 KO mice (Figure 3C). These data suggest that the presence of Pin1 promotes M1-polarized differentiation of macrophages accompanied by increases in *Gmcsf* expression. M1 or M2 macrophage differentiation is regulated by cytokines released from T cells and tissues. Conversely, T cells are activated by pro-inflammatory cytokines produced by macrophages.

Th17 cells are abundant in the normal state in the gut. DC cells and macrophages produce cytokines IL-6, IL-23 and IL-1β and then drive Th17 cells to produce IL-17 [17]. Secreted IL-17 induces the recruitment and activation of neutrophils as well as the production of anti-microbial peptides from epithelial cells [18,19]. Excessive Th17-induced responses have been implicated in the development of IBD [20,21]. Hence, we investigated expression levels of the mRNA of *Il23a* and *Il17a*. DSS treatment markedly increased both mRNA levels in WT mice, and these elevations were significantly suppressed in Pin1 KO mice (Figure 3D).

## 4. Therapeutic Effects of the Pin1 Inhibitor Juglone on DSS-Induced Colitis in Mice

Finally, we investigated the effects of orally administered Pin1 inhibitor, Juglone, on the development of DSS-induced colitis in mice. There were no significant differences in weight loss or colon length between mice with versus without daily oral Juglone (5 mg/Kg BW) administration (Figure 4A). However, Juglone treatment significantly ameliorated DSS-induced tissue damage in the colon (Figure 4B). In addition, DSS-induced elevations in *Il6* and *Il1b* mRNA and cleaved caspase 3 levels were significantly suppressed by Juglone treatment (Figure 4C,D). Juglone had no effect on the normalization of *Tnfa* mRNA expression (Figure 4D). These results suggest that the Pin1 inhibitor exerts therapeutic effects on DSS-induced colitis in mice, raising the possibility of Pin1 serving as a novel therapeutic target for IBD.

## 5. Discussion

To date, many physiological as well as pathogenic roles of Pin1 have been elucidated. For example, Pin1 interacts with and promotes the degradation of tau, advantageously contributing to protection from Alzheimer disease [12]. As an unfavorable function, in contrast, Pin1 overexpression is observed in most cancers and its levels correlate with the aggressiveness of a malignancy and a poor prognosis [10,11]. Several recent reports have unraveled the contribution of Pin1 to the inflammatory and fibrotic changes in various tissues such as the liver and kidney [15,22].

This is the first study demonstrating the pathological role of Pin1 in the development of DSS-induced colitis. Pin1 expression was shown to be markedly increased in the DSS-treated mouse colons. In addition, global *Pin1* KO mice showed resistance to DSS-induced colitis development characterized by body weight loss, colon length reduction and epithelial tissue disruption.

Interestingly, ablation of the Pin1 gene reduced expressions of pro-inflammatory cytokines including Th17-related cytokines and increased the M2 macrophage population in the colons of DSS-treated mice. Many functions of Pin1 in the regulation of immune or inflammatory responses were previously described in the literature [23,24]. T helper cells reportedly play a key role in the complex pathogenesis of IBD [25,26]. In particular, Th17 cells which are abundant in the mucous layer are differentiated from naive T cells in response to stimulation with IL-6 and TGF-β, and then produce IL-17 [27,28]. During IBD development, differentiation into Th17 cells is enhanced, and an increase in the GM-CSF concentration further aggravates inflammation by increasing blood cell infiltration and pro-inflammatory cytokine expressions, which in turn mediate IL-23 production by dendritic cells [17,29].

Besides the involvement of Th17 cells, multiple mechanisms underlying the contribution of hematopoietic Pin1 to the pathogenesis of colitis are possible. First, Pin1 interacts with and controls the ARE mRNA-binding activity of AUF1 in T lymphocytes and eosinophils. AUF1 induces the catabolization of several mRNAs including those of *Gmcsf* and *Tgfβ* by preventing exosome-mediated translation. Pin1 reportedly induces AUF1 isomerization, leading to the loss of its binding ability with *Gmcsf* mRNA, and thereby stabilizing *Gmcsf* mRNA [13,30]. 

In addition, Pin1 binds to the p65 subunit of NF-κB and promotes its transcriptional activity [10]. NF-κB regulates the expressions of many pro-inflammatory mediators, including TNFα. TNFα propagates the inflammatory response via the transmigration of neutrophils from the vascular space into tissues including the hepatic parenchyma and lamina propria. These activated neutrophils then directly injure epithelial cells and vascular endothelial cells by releasing oxidants and proteases. Furthermore, Pin1 acts as a molecular switch for TNFα-induced priming of NADPH oxidase, leading to free radical production [23]. 

Taken together, these findings show that orally-administered Pin1 inhibitor exerts a significant protective effect against DSS-induced colitis, raising the possibility of Pin1 inhibitors serving as novel therapeutic agents for treating IBDs. We conclude that Pin1 plays a key role in the pathogenesis of DSS-induced colitis in mice. Although further studies are necessary to fully elucidate the roles of Pin1 in colitis development, the development of new agents with high specificity for inhibiting Pin1 activity are eagerly awaited.

## Figures and Tables

**Figure 1 cells-10-01230-f001:**
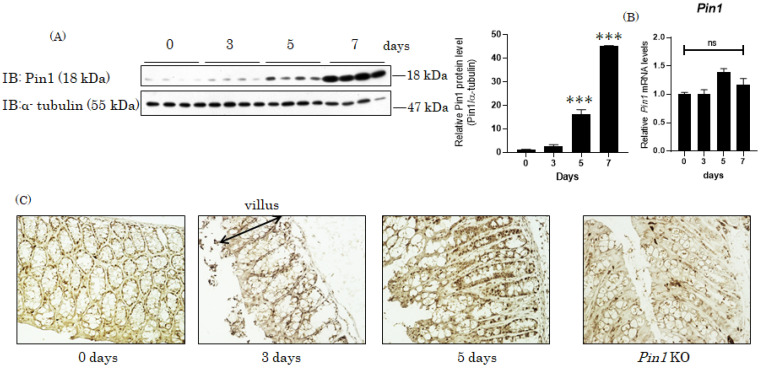
Pin1 expression was markedly increased in mice with DSS-induced colitis. (**A**,**B**) The colons were isolated from WT mice treated with drinking water containing DSS for 3, 5 and 7 days (n = 5–6), and Pin1 protein (a) and mRNA (b) levels were determined by immunoblotting and real time -PCR (RT-PCR), respectively. The ratio of Pin1 protein/ tubulin is presented as a bar graph (right panel of a). The *Y* axis of (b) represents ΔΔCT. Significant differences were identified by applying a one-way ANOVA followed by Tukey’s multiple comparisons test. ns, not significant, *** *p* < 0.001 Values are means + SEM. (**C**) Colonic samples isolated from WT mice treated with drinking water containing DSS for 0, 3 and 5 days and global *Pin1* KO mice without treatment were stained immunohistochemically.

**Figure 2 cells-10-01230-f002:**
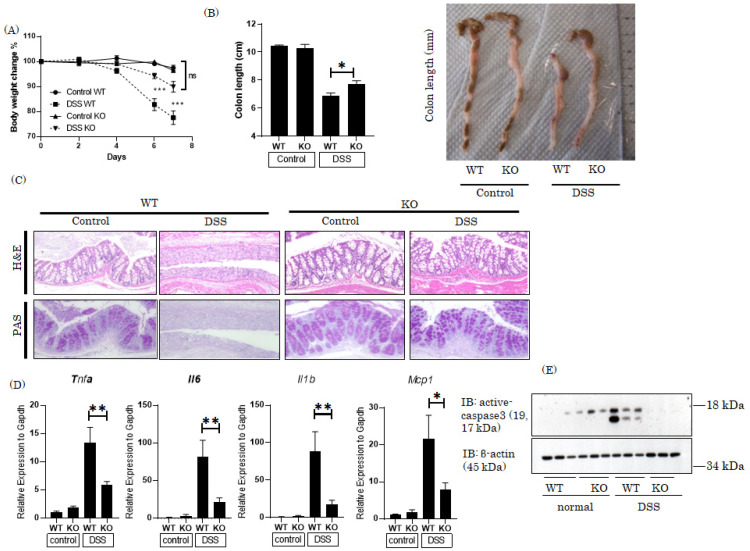
Global *Pin1* KO mice were resistant to the development of DSS-induced colitis. Global *Pin1* KO mice were generated by mating *Pin1* fl/fl and CAG cre-TG mice (n = 5–8). *Pin1* KO and control *Pin1* fl/fl mice were treated with 3% DSS for 7 days. (**A**) Body weight, (**B**) colon length (**C**) and histology based on H&E and PAS staining of colonic sections (**D**) The mRNA levels of *T**nfα*, *Il6*, *Il1b* and *Mcp1* were measured by RT-PCR. (**E**) Active forms of caspase 3 and α-tubulin in the colons were detected by immunoblotting. Significant differences were identified by applying a one-way ANOVA followed by Tukey’s multiple comparisons test. ns, not significant, * *p* < 0.05, ** *p* < 0.01, *** *p* < 0.001 Values are means ± SEM.

**Figure 3 cells-10-01230-f003:**
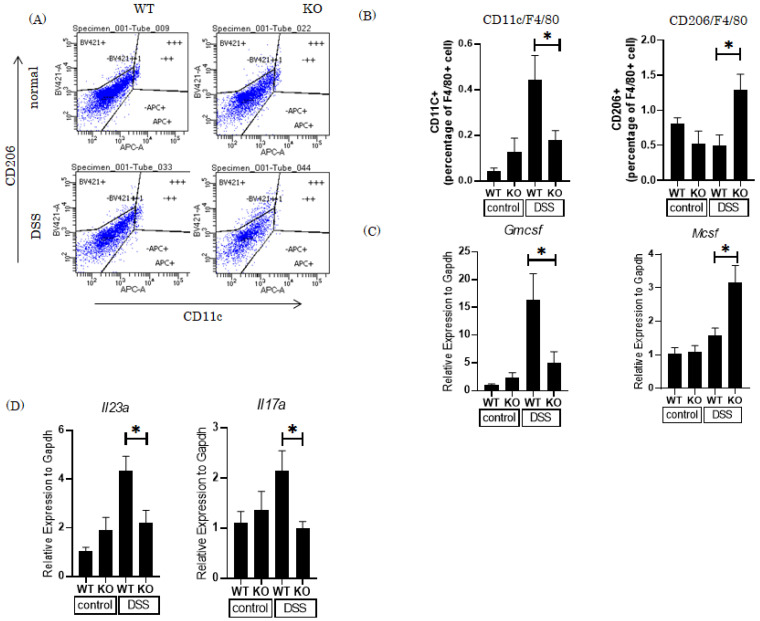
Increased M2 macrophages and suppressed Th17-related cytokines and antimicrobial peptides in the Pin1 KO mouse colon tissues after DSS treatment. Global *Pin1* KO mice were generated by mating *Pin1* fl/fl and CAG cre-TG mice. (n = 5–8) (**A**) Representative FACS plots show the percentage of M1 and M2 macrophages among total macrophages isolated from the colonic lamina propria of *Pin1* KO and control *Pin1* fl/fl mice that had been treated with 3% DSS for 7 days. (**B**) The percentages of M1 and M2 macrophages. (**C**) The mRNA expression levels of *Gmscf* and *Mcsf* in colonic tissues isolated from *Pin1* KO and control Pin1fl/fl mice that had been treated with 3% DSS for 7 days. (**D**) The mRNA expression levels of *Il23a* and *Il17* in colonic tissues isolated from *Pin1* KO and control *Pin1* fl/fl mice that had been treated with 3% DSS for 7 days. Significant differences were identified by applying a one-way ANOVA followed by Tukey’s multiple comparisons test. * *p* < 0.05, Values are means ± SEM.

**Figure 4 cells-10-01230-f004:**
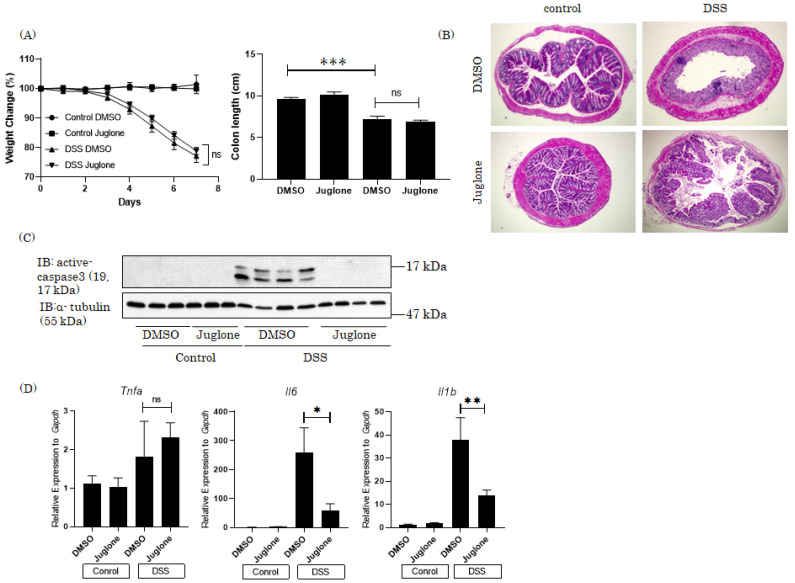
Pin1 inhibitors blocked the development of DSS-induced colitis. Juglone or solvent DMSO alone was administered orally once a day to the mice given drinking water with or without 3% DSS for 7 days (n = 6–10). (**A**) Body weight change of each mouse (**B**) Histology based on H&E staining of the colonic sections. (**C**) Active forms of caspase 3 and α-tubulin in the colons were detected by immunoblotting. (**D**) The mRNA expression levels of *TNFα*, *Il6* and *Il1b* in the colons were measured by RT-PCR. Significant differences were identified by applying a one-way ANOVA followed by Tukey’s multiple comparisons test. n.s.: not significant, * *p* < 0.05, ** *p* < 0.01, *** *p* < 0.001 Values are means ± SEM.

**Table 1 cells-10-01230-t001:** Primer sequences.

	Forward	Tm	Reverse	Tm	Product Size
*Gapdh*	TGATGGGTGTGAACCACGAG	63	GGGCCATCCACAGTCTTCTG	63	178
*Tnfa*	GAACTGGCAGAAGAGGCACT	63	AGGGTCTGGGCCATAGAACT	63	163
*Mcp1*	TGGTCCCTGTCATGCTTCTG	63	TCTGGACCCATTCCTTCTTG	63	208
*Il6*	AGTTGCCTTCTTGGGACTGA	63	CAGAATTGCCATTGCACAAC	63	151
*Il17a*	TCGAGAAGATGCTGGTGGGT	58.4	CTCTGTTTAGGCTGCCTGGC	60.4	71
*Il23a*	GGTGGCTCAGGGAAATGT	55.8	GACAGAGCAGGCAGGTACAG	60.4	67
*Gmcsf*	ATGCCTGTCACGTTGAAT GAAG	63	GCGGGTCTGCACAGATGTTA	63	81
*Pin1*	CGGCAGGAAAAGATCACCAG	63	TCCCCTGTCCGTAGAGCAAA	63	175
*Il1b*	CGTGGACCTTCCAGGATGAG	63	GCTCATATGGGTCCGACAGC	63	146

## Data Availability

The data presented in this study are available on reasonable request.

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
