# Peer review of "Pathological Role of Pin1 in the Development of DSS-Induced Colitis"

_cells, 2021, doi:10.3390/cells10051230_

Round 1

Reviewer 1 Report

In Yasuka Matsunaga et al., the authors performed the pathological role of Pin1 in the development of DSS-induced colitis. Even though the study seems interesting at first sight, there are many issues that hinder abetter assessment of its value at this moment.

Materials and methods: the methodology lacks a lot of key information and many references to the literature.

Antibodies

- please complete such details as: dilutions, catalog numbers, RRID number, company name, city, country etc.

Real-time PCR

-total RNA isolation methodology - what was the RNA quality / integrity?

- how much RNA has been transcribed into cDNA - please fill in the details of digesting samples with DNAse

- please provide the primer sequences in the form of a table + Tm + product size.

Western Blotting

-how many µg of protein was there in the sample?

“Samples were electrophoresed with SDS (sodium dodecyl sulfate)-polyacrylamide gel, transferred to polyvinylidene difluoride membranes, and subjected to immunoblotting using Supersignal West Pico Substrate” - was there an antibody incubation step? Were the samples normalized to the reference gene?

Results:

Fig. 1

- please explain where is the protein size marker? How did the authors identify the proteins, since they are absent?

- the molecular mass are missing

  1. B) - scales are not properly described (e.g. what is the unit?)

- was mRNA expression performed by RT-PCR (reverse transcription-PCR) or by qPCR?

- do the results represent the mean ± SEM or SD?

  1. C) histology

- descriptions, marked anatomical structures, lesion markers, negative / positive controls are missing.

- are the results were done in biological replicates? 

Statistical analysis

-please explain why the t-test was used? The one-way anova analysis of variance seems more appropriate. What were the groups? How many technical and how many biological repetitions? Which results are considered statistically significant?

“These experiments were approved by Hiroshima University Animal Research 89 Committee” - please complete the consent number of the ethics committee.

Author Response

In Yasuka Matsunaga et al., the authors performed the pathological role of Pin1 in the development of DSS-induced colitis. Even though the study seems interesting at first sight, there are many issues that hinder a better assessment of its value at this moment.

â‘ Materials and methods: the methodology lacks a lot of key information and many references to the literature.

Antibodies

- please complete such details as: dilutions, catalog numbers, RRID number, company name, city, country etc.

→We added the requested information to the Methods section.

Real-time PCR

-total RNA isolation methodology - what was the RNA quality / integrity?

→We appreciate the reviewer’s suggestions. We have added the following information to the revised manuscript.

“We used an Oligotex™ -dT30 mRNA Purification Kit to isolate mRNA from total RNA obtained with Sepazol. For every sample, the 260/280 Absorbance Ratio was near 2.0. “

- how much RNA has been transcribed into cDNA

→50 ng of mRNA were reverse-transcribed into cDNA. We included this information in the Methods section.

 - please fill in the details of digesting samples with DNAse

→We added the following sentences to the Methods section.

“We used an mRNA Purification Kit and did not perform DNAase treatment. Then, we employed a verso cDNA reverse-transcribed synthesis kit. This kit includes an RT enhancer which prevents the contamination of genomic DNA.”

- please provide the primer sequences in the form of a table + Tm + product size.

→The attached table 1 contains this information.

Western Blotting

-how many µg of protein was there in the sample?

→We used 40 ug of protein for each sample for western blotting.

“Samples were electrophoresed with SDS (sodium dodecyl sulfate)-polyacrylamide gel, transferred to polyvinylidene difluoride membranes, and subjected to immunoblotting using Supersignal West Pico Substrate” - was there an antibody incubation step? Were the samples normalized to the reference gene?

→We appreciate your pointing this out. We have corrected it in the revised Materials and Methods section. The following sentences were added to the Methods section.

“The same amounts of samples (40 ug) were electrophoresed with SDS (sodium dodecyl sulfate)-polyacrylamide gel, and then transferred to polyvinylidene difluoride membranes. The membranes were blocked with 3% bovine serum albumin for 1hr at room temperature, and then incubated with primary antibody (1:3000). After being washed with PBS containing 0.1% Tween 20 three times, the membranes were reacted with HRP-conjugated secondary antibody (1:5000) for 1 hr at room temperature. After washing, proteins on the membranes were subjected to immunoblotting using Supersignal West Pico Substrate (Thermo Scientific, Waltham, MA, USA).”

Results:

Fig. 1

- please explain where is the protein size marker? How did the authors identify the proteins, since they are absent?

→We apologize for carelessly neglecting to include the size of the protein size marker. We have added it to the revised figure.

- the molecular masses are missing 

→Thank you for pointing this out. We have added the molecular mass to the revised figure. 

  1. B) - scales are not properly described (e.g. what is the unit?)

→We apologize for the confusion. We now use Relative Pin1 mRNA levels to avoid misunderstanding.

- was mRNA expression performed by RT-PCR (reverse transcription-PCR) or by qPCR?

→RT-PCR means real time-PCR. Therefore, Figure 1B shows qPCR results. We have corrected this in figure 1 legend.

- do the results represent the mean ± SEM or SD?

→We show all data as means+S.E.M. This is noted in the legend of figure1.

  1. C) histology

- descriptions, marked anatomical structures, lesion markers, negative / positive controls are missing.

→Thank you for pointing this out. We have added Pin1 ko mouse histology as a negative control and marked the anatomical structures in the revised figure. The avidin-biotin complex (ABC) technique was used for this immunohistochemical analysis. Therefore, it is not possible to stain with another antibody for lesion markers. We have corrected this in revised Methods section.

- are the results were done in biological replicates? 

→Yes. We confirmed the replicate experiments.

Statistical analysis

-please explain why the t-test was used? The one-way anova analysis of variance seems more appropriate. What were the groups? How many technical and how many biological repetitions? Which results are considered statistically significant?

We apologize for the confusion. We used the one-way Anova analysis instead of the t-test. We performed at least two independent experiments. We now indicate the significant differences.

“These experiments were approved by Hiroshima University Animal Research  Committee” - please complete the consent number of the ethics committee.

 →We added this information to the Methods section.

Reviewer 2 Report

Inflammatory bowel diseases (IBDs) display a severe health burden to diseased individuals and are expected to rise in prevalence. Patients have to cope with several co-morbidities and, therefore, it is essential to elucidate underlying molecular mechanisms in order to develop therapeutic strategies. In their study, the authors aimed to elucidate the role of Peptidyl-prolyl cis-trans isomerase NIMA-interacting 1 (PIN1) in colitis. For this purpose, they have used a comprehensive approach by inducing DSS-mediated colitis in wild type, global Pin1 knockout mice and, in addition, treated mice with a PIN1 inhibitor. Their findings on this topic indicate the relevance of PIN1 in colitis and will be of interest to the readership of Cells. A number of major and minor points should be addressed prior to publication as detailed below.

Major

The inflammation-associated parameters determined after Juglone treatment (Figure 4) were different from the previous data on DSS treatment (Figure 2). In order to be able to compare the effects shown in these two figures, similar parameters should be shown.

The authors describe that “After being given drinking water supplemented with 3% DSS for 7 days, all mice showed symptoms including diarrhea, rectal bleeding, reduction in colon length, body weight loss and decreased activity mimicking the clinical and histological features of IBDs. Microscopic observation indicated severe damage in the epithelial layer.” (line 152). Corresponding data should be included in the manuscript or supplement.

Figure 1 demonstrates that PIN1 protein levels increase during intestinal inflammation and that Pin1 gene expression increases, too, when normalized to tubulin. How do the authors explain that results were not significant when normalizing to Gapdh? Was GAPDH upregulated during colitis on protein or mRNA levels similar to Pin1? Please show corresponding data. Moreover, the IHC staining should be done for day 0, 3, 5 and 7.

Minor

Please show the knockout efficiency in global Pin1 knockout mice by IHC/western blot and qRT-PCR for PIN1.

The catalog numbers of antibodies should be provided.

Please elaborate on the experimental steps to perform hematoxylin and eosin (H&E) and Periodic acid-Schiff (PAS) staining or add corresponding references.

“Whole-body Pin1 KO mice” should be referred to as “Global Pin1 knockout mice”.

The authors did not pay attention to the nomenclature. Murine gene names should be written in small letters (only the first letter is capitalized) and be italicized. Murine proteins are written in capital letters.

Author Response

Inflammatory bowel diseases (IBDs) display a severe health burden to diseased individuals and are expected to rise in prevalence. Patients have to cope with several co-morbidities and, therefore, it is essential to elucidate underlying molecular mechanisms in order to develop therapeutic strategies. In their study, the authors aimed to elucidate the role of Peptidyl-prolyl cis-trans isomerase NIMA-interacting 1 (PIN1) in colitis. For this purpose, they have used a comprehensive approach by inducing DSS-mediated colitis in wild type, global Pin1 knockout mice and, in addition, treated mice with a PIN1 inhibitor. Their findings on this topic indicate the relevance of PIN1 in colitis and will be of interest to the readership of Cells. A number of major and minor points should be addressed prior to publication as detailed below.

Major

The inflammation-associated parameters determined after Juglone treatment (Figure 4) were different from the previous data on DSS treatment (Figure 2). In order to be able to compare the effects shown in these two figures, similar parameters should be shown.

→We appreciate your pointing this out. Juglone treatment had no effect on weight loss or colon length. Moreover, there was no significant difference in Tnfa mRNA expression between the juglone-treated and control mice. We have added these data to the revised figure 4.

The authors describe that “After being given drinking water supplemented with 3% DSS for 7 days, all mice showed symptoms including diarrhea, rectal bleeding, reduction in colon length, body weight loss and decreased activity mimicking the clinical and histological features of IBDs. Microscopic observation indicated severe damage in the epithelial layer.” (line 152). Corresponding data should be included in the manuscript or supplement.

→We appreciate this suggestion. This is general knowledge. We have modified sentences in the revised paper accordingly.  

Figure 1 demonstrates that PIN1 protein levels increase during intestinal inflammation and that Pin1 gene expression increases, too, when normalized to tubulin. How do the authors explain that results were not significant when normalizing to Gapdh? Was GAPDH upregulated during colitis on protein or mRNA levels similar to Pin1? Please show corresponding data. Moreover, the IHC staining should be done for day 0, 3, 5 and 7.

 →We appreciate the reviewer’s advice. The bar graph in Fig.1A shows the relative Pin1 protein levels, normalized to tubulin, while Fig.1B reveals relative Pin1 mRNA levels normalized to Gapdh. To avoid misunderstanding, we revised this panel. Furthermore, we have added IHC staining for days 0, 3 and 5. However, a full comparison to mice with DSS treatment for 7 days is not feasible because DSS treatment for 7 days causes severe damage and alters morphology (Figure 2C).

Minor

Please show the knockout efficiency in global Pin1 knockout mice by IHC/western blot and qRT-PCR for PIN1.

→Thank you for pointing this out. we have added knockout efficiency to the revised supplemental figure1. 

The catalog numbers of antibodies should be provided.

→We added this information to the revised Methods section.

Please elaborate on the experimental steps to perform hematoxylin and eosin (H&E) and Periodic acid-Schiff (PAS) staining or add corresponding references.

→We added the details of the method used to the revised manuscript.

“Whole-body Pin1 KO mice” should be referred to as “Global Pin1 knockout mice”.

-→We appreciate the reviewer’s suggestion. We have made the correction in the revised manuscript.

The authors did not pay attention to the nomenclature. Murine gene names should be written in small letters (only the first letter is capitalized) and be italicized. Murine proteins are written in capital letters.

-

We apologize for these mistakes, which have been corrected.

Round 2

Reviewer 1 Report

There are no comments.

Reviewer 2 Report

All previous concerns have been addressed.